# Emergence and Genomic Characterization of Multidrug Resistant *Candida auris* in Nigeria, West Africa

**DOI:** 10.3390/jof8080787

**Published:** 2022-07-27

**Authors:** Rita Oladele, Jessica N. Uwanibe, Idowu B. Olawoye, Abdul-Wahab O. Ettu, Jacques F. Meis, Christian T. Happi

**Affiliations:** 1College of Medicine, University of Lagos, Lagos 102212, Nigeria; drritaoladele@yahoo.com; 2Department of Biological Sciences, Redeemer’s University, Ede, Osun 232101, Nigeria; uwanibej@run.edu.ng (J.N.U.); olawoyei0303@run.edu.ng (I.B.O.); 3African Centre of Excellence for Genomics of Infectious Diseases, Ede, Osun 232101, Nigeria; 4Lagos State Health Service Commission, Lagos 102273, Nigeria; omoope@gmail.com; 5Marigold Hospital and Critical Care Centre, Lagos 101241, Nigeria; 6Department of Medical Microbiology and Infectious Diseases (C70), Canisius Wilhelmina Hospital (CWZ), 6532 SZ Nijmegen, The Netherlands; j.meis@cwz.nl; 7Centre of Expertise in Mycology Radboudumc/CWZ, 6532 SZ Nijmegen, The Netherlands

**Keywords:** *Candida auris*, multidrug resistance, azoles, sequencing, fungaemia

## Abstract

*Candida auris* is an emerging multidrug-resistant fungal pathogen that has become a worldwide public health threat due to the limitations of treatment options, difficulty in diagnosis, and its potential for clonal transmission. Four ICU patients from three different healthcare facilities in Southern Nigeria presented features suggestive of severe sepsis and the blood cultures yielded the growth of *Candida* spp., which was identified using VITEK 2 as *C. auris*. Further confirmation was performed using whole genome sequencing (WGS). From the genomic analysis, two had mutations that conferred resistance to the antifungal azole group and other non-synonymous mutations in hotspot genes, such as ERG2, ERG11, and FKS1. From the phylogenetic analysis, cases 2 and 4 had a confirmed mutation (*ERG11:*Y132F) that conferred drug resistance to azoles clustered with clade 1, whilst cases 1 and 3 clustered with clade 4. Three of the patients died, and the fourth was most likely a case of colonization since he received no antifungals and was discharged home. These first cases of *C. auris* reported from Nigeria were most likely introduced from different sources. It is of public health importance as it highlights diagnostic gaps in our setting and the need for active disease surveillance in the region.

## 1. Introduction

*Candida* species are the predominant cause of nosocomial fungal infections, causing bloodstream infections (BSI) with significant mortality [1], therefore eliciting a major threat to intensive care unit (ICU) patients [2,3]. *Candida auris* is an emerging multidrug-resistant fungal pathogen that has become a worldwide public health threat due to limitations of treatment options based on antifungal resistance issues and its potential for clonal transmission. In addition, difficulties in identification using conventional phenotypic and molecular techniques, the unknown population prevalence, the uncertain environmental niches, and the unclear mechanisms of spread have hindered control [4]. Molecular epidemiological investigations of *C. auris* outbreaks generally show clusters of highly related isolates, supporting local and ongoing transmission. The analysis of outbreaks and individual cases has also revealed genetic complexity, with isolates from different clades detected in Germany, the United Kingdom, and the United States, suggesting multiple introductions into these countries, followed by local transmission [5]. Since it was first isolated from an ear canal sample of a Japanese patient in 2009, it has been isolated in several countries on five continents across the world. Mortality rates have varied significantly among geographic regions from Asia, the far East, and the United States have reported a 50% mortality rate; this is in contrast to Venezuela, where the 30-day survival rate was 72% [6]. In Africa, *C. auris* has been previously reported in South Africa, Kenya, Sudan, Egypt, and Algeria [7]. In Nigeria, the prevalence of *C. auris* was unknown because no cases of colonization or infection had been reported before [8]. However, given its geographical location, with frequent travel of the population for business and medical purposes to India, South Africa, the USA, and the UK to the west, the country is at high risk for the spread of this pathogen. 

In this study, we report the first four cases of *C. auris* infection and colonization in Nigeria. This is also the first report of a *C. auris* outbreak in West Africa, to the best of our knowledge.

### 1.1. The Cases

Over a three-month period, four cases of *Candida* bloodstream infections were reported from four different facilities in Nigeria: two from ICUs in two private facilities in Lagos, one from a tertiary hospital in Ibadan (which is approx. 200 km from Lagos), and the last, from a tertiary hospital in Lagos. The isolates were initially identified phenotypically as non-*Candida albicans*. However, with the deteriorating clinical conditions of the patients, the isolates were sent to the Lagos University Teaching Hospital (LUTH), which had VITEK 2 (9.01 version) and yeast cards routinely available for species identification and antifungal susceptibility testing with breakpoints defined based on the CDC guidance for *C. auris* minimum inhibitory concentration (MIC) interpretation [9]. The organisms were identified as *C. auris*. Table 1 below is a summary of patient demographics, clinical presentation, diagnosis, and outcome.

#### 1.1.1. Case 1

A 60-year-old known type 2 diabetic, a hypertensive male who had defaulted on clinic attendance for 15 years, presented at a private facility with a 2-month history of Diabetic Foot Syndrome in his right foot. The patient had intermittent pyrexia. He had been treated with various antibiotics, including ceftriaxone, amoxicillin + clavulanic acid, and levofloxacin. A lower-extremity arterial doppler showed plaques in the lower limb arteries with 51% stenosis. A computed tomography (CT) angiogram of the lower limbs also revealed extensive artery disease (extensive luminal narrowing) in the lower limbs’ arteries. An X-ray of the right foot showed features of osteomyelitis of the navicular bone. The total leucocyte counts were 17,100/cc, and the fasting blood glucose was 181 mg/dL. Sepsis persisted, and he was admitted into the high dependency unit (HDU) and subsequently moved to the intensive care unit (ICU). A chest X-ray performed on the ninth day of ICU admission showed features of cardiomegaly with bilateral pleural effusion. On the twelfth day of ICU admission, the patient had an above-the-knee surgical amputation of the right lower limb. A blood culture taken on the same day grew Extended-spectrum beta-lactamases (ESBL) producing carbapenem-resistant *Klebsiella pneumoniae* sensitive to tigecycline and polymyxins, resistant to ciprofloxacin, gentamicin, cotrimoxazole, and amikacin. The sputum cultures on the 16th day yielded a profuse growth of *Candida* spp. Urine cultures also grew *Candida* spp. By the 26th day, a second blood culture was taken from a tunneled catheter and central line grew *Candida* spp. The *Candida* spp. was later (Mycology laboratory in a tertiary hospital) identified as *C. auris.*

The patient was treated with an empirical broad-spectrum antibiotic regimen of levofloxacin, metronidazole, ceftriaxone, and clarithromycin prior to the initial blood culture, which was then changed to vancomycin, meropenem, and polymixin B. Oral voriconazole 200 mg every 12 h was commenced prior to the collection of the repeat blood culture and the dose was subsequently reduced to 200 mg daily with decreasing renal function. However, his clinical condition deteriorated post-haemodialysis on the 42nd day of admission, where he had a cardio-respiratory arrest with the eventual demise of the patient after several attempts at resuscitation.

#### 1.1.2. Case 2

A 67-year-old known type 2 diabetic, a hypertensive female with associated hypothyroidism, following treatment (total thyroidectomy 6 years ago) for thyrotoxicosis, presented in a private ICU facility following a referral from another private facility with a history of tiredness for 3 days, loss of consciousness, and difficulty breathing for a duration of 8 h. There was no history of chest pain, orthopnoea, or bilateral leg swelling. The patient used amlodipine, metformin, levothyroxine, and bendroflumethiazide for her chronic diseases, as well as eyedrops timolol, bimatoprost, and dorzolamide following trabeculectomy/cataract surgery (7 years ago).

Further examination revealed a GCS of 12/15, an SPO_2_ of 40% on room air with a respiratory rate of 40 cpm, bilateral coarse crepitations, and a heart rate of 100 bpm. A diagnosis of severe sepsis was made with a focus on the chest infection with a differential diagnosis of COVID-19. Investigations revealed raised total leucocyte counts (15,000/cc), random blood glucose of 137 mg/dL, and a positive COVID-19 PCR test. A blood culture taken 18 h after admission grew a yeast that was initially identified as *Candida* spp. but was later identified as *C. auris*. 

The patient was treated with an empirical broad-spectrum antibiotic regimen of ceftriaxone, metronidazole, azithromycin, and amphotericin B deoxycholate. She was mechanically ventilated for 4 days and successfully weaned off of the ventilator, and subsequently discharged with a rebreathing mask at 15 L/min to the private hospital on day 6 of admission. However, the patient died a week later from severe sepsis.

#### 1.1.3. Case 3

The patient was a 74-year-old male being treated for prostate cancer at a tertiary hospital in the city of Ibadan, Nigeria. He was admitted for a prostatectomy (resection of tumor) as part of his treatment protocol. The patient received empirical meropenem and vancomycin. Postoperative, he developed sepsis with a focus on the urinary tract (urinary catheter present). The collected urine culture yielded a moderate growth of yeast cells that were identified as *Candida* spp. The patient received no antifungal treatment. Blood culture was taken concurrently and also grew yeast cells, which were initially identified as *Candida* spp. but later identified as *C. auris*. The patient was discharged home following the resolution of symptoms and continues treatment at an oncology department, where he is presently undergoing chemotherapy. The blood culture yield of *C. auris* was most likely a case of colonization.

#### 1.1.4. Case 4

The patient was a 48-year-old female that was being managed for systemic lupus erythematosus (SLE) at the Lagos University Teaching Hospital. She had been on the ward for 8 weeks. The patient had been managed for nosocomial sepsis at different times during admission. She was treated with different courses of broad-spectrum antibiotics (ceftriaxone, metronidazole, piperacillin/tazobactam, levofloxacin, meropenem, gentamicin, amikacin, and clotrimazole) and fluconazole during the periods of these hospital-acquired infections. She intermittently received oxygen via a face mask (respiratory support) and had invasive devices in place (central venous line, urinary catheters, etc.). On day 50 of admission, she yet again developed symptoms and signs of severe sepsis and was unresponsive to empirical meropenem followed by tigecycline. With decreasing renal function, she commenced hemodialysis on day 58. Blood cultures grew *C. auris*. The patient was given intravenous voriconazole 200 mg b.d, but her clinical condition continued to deteriorate and was, therefore, transferred to the ICU on day 64 for ventilatory support but died of severe sepsis on day 71.

## 2. Materials and Methods

### 2.1. DNA Extraction and Library Preparation

The genomic DNA from the cultured isolates was extracted using the Plant/Fungi DNA Isolation Kit (Norgen Biotek CORP, Thorold, ON, Canada). The DNA samples were quantified using a Qubit fluorometer (ThermoFisher Scientific, Waltham, MA, USA) using a dsDNA High Sensitivity Assay. The sequencing libraries were prepared using the Nextera DNA Flex Preparation Kit (Illumina, San Diego, CA, USA) adopted from a previous study [10]. 

### 2.2. Whole Genome Sequencing and Bioinformatics Analysis

Four isolates were sequenced on the Illumina MiSeq using an Illumina DNA flex library preparation kit and the v2 300 cycle kit. FASTQ files that were generated from the MiSeq underwent quality control checks using FASTQC and further improved with fastp [11]. The resulting FASTQ paired-end files were mapped against the *C. auris* reference genome (GCF_002775015.1), which is 12.7 million base pairs (Mbp) long and contains seven chromosomes using BWA MEM [12] algorithm. Duplicate reads were marked and removed from the resulting mapped reads using SAMtools, and a coverage plot was generated by normalizing the maximum read depth at 14,964 base pair window (15 kbp).

Variants were called using Freebayes [13] v1.3.2 by using the variant filtering parameters of the Mean mapping quality (MQM) > 45 and Read depth (DP) > 10. The VCF file was annotated with snpEff [14] v 5.0e and filtered using SnpSift [15] v5.0e to focus on specific genes of interest, *ERG11*, *ERG3*, *ERG2*, *ERG6*, *FKS1*, *FKS2,* and *FUR1,* with nonsynonymous mutations excluding mutations in upstream, downstream, and intergenic regions.

### 2.3. Phylogenetic Analysis

We analyzed 37 *C. auris* whole genomes previously studied [5], which have already been classified into four distinct lineages, in addition to the four isolates from this study. The VCFs generated from variant calls were merged with BCFtools [16] v1.10.2, taking only SNPs into account. Genome-wide SNP phylogenetic analysis was performed on all samples by selecting unambiguous SNPs (n = 174,758) with an ultrafast bootstrap value of 1000 and a generalized time-reversible model using an IQ-TREE [17], which was visualized on a figtree.

## 3. Results

### 3.1. Antifungal Susceptibility Test (AFST)

To examine the resistance levels to the available drugs, antifungal susceptibility tests were performed using fluconazole, voriconazole, posaconazole, amphotericin B, caspofungin, micafungin, and anidulafungin. The results exhibited a high minimum inhibitory concentration (MIC) to triazoles in cases 1, 2, and 4 and susceptibility to fluconazole in case 3 (Table 2).

### 3.2. Genome Analysis

The average coverage of *C. auris* reads that mapped against the reference genome after quality improvement with fastp were 44.15×, 51.85×, 59.23× and 67.26× respectively with all seven chromosomes sequenced as seen in Figure 1.

### 3.3. Variant Analysis

Following genome characterization and assembly, specific mutations conferring antifungal resistance were investigated in hotspot genes such as *ERG2*, *ERG3*, *ERG6, ERG11*, *FKS1*, *FKS2*, and *FUR1*. Non-synonymous mutations were detected in the *ERG11* gene of all isolates, with case 2 having additional non-synonymous mutations (L148I, R937S, I701V, and I694V) in the *FKS1* gene and case 4 with another non-synonymous mutation (E39D) in the *ERG2* gene (Table 3). 

### 3.4. Phylogenetic Analysis

From the phylogenetic analysis, cases 2 and 4, had confirmed mutations (*ERG11:*Y132F) that confer drug resistance to azoles clustered with clade 1, whilst cases 1 and 3 clustered with clade 4 (Figure 2).

## 4. Discussion

In this study, we report the first cases of *C. auris* in Nigeria, which also happen to be the first reported cases in the West African region. Antifungal susceptibility tests, in addition to whole genome sequence analysis, revealed high resistance levels to triazoles, as the Tyr123Phe (Y132F) substitution mutation in the *ERG11* gene has been previously studied and known to elicit fluconazole and voriconazole resistance.

*C. auris* causes outbreaks in hospitals, and cases have been reported in Colombia, Venezuela, Israel, South Africa, India, and the United Kingdom [18,19,20,21,22,23]. Within weeks of an admitted index case, patients in a long-term care facility can be colonized with *C. auris* [22]. It has been identified among patients at long-term acute care hospitals and likewise among those who have received healthcare in countries with increasing *C. auris* transmission, or being immunocompromised, received recent broad-spectrum antibiotics or antifungals, or had a central venous catheter [23,24]. It can be misidentified as *Candida haemulonii*, *Candida famata*, *Saccharomyces cerevisiae*, or *Rhodotorula glutinis* with the analytical profile index strips or VITEK 2 [25,26].

As part of this study, 600 *Candida* isolates from different sites in Nigeria in 2019 were sent for sequencing in search of possible *C. auris*, 210 were successfully sequenced, and none were *C. auris*. In discussion with collaborators at CDC Atlanta, information of a case of *C. auris* isolated from a Nigerian child who was in New York for Medical care flagged the alarm of the possibility of the infection being acquired from Nigeria. This was at the peak of the first wave of COVID-19; thus, investigating it was not feasible. Here, we report four recent cases of *C. auris* observed in Nigeria over a three-month period. Previous genomic studies have revealed that the four major distinct clades of *C. auris* emerged independently at different geographical points across the world [27], and our phylogenetic analysis implies that there might have been at least two different introductions of *C. auris* in Nigeria that emerged independently of each other. Clade 1 is the most globally spread *C. auris* and has been found in many countries, such as Canada, France, Germany, India, Kenya, Pakistan, Saudi Arabia, the United Kingdom, UAE, and the USA. However, clade 4 is mostly found in the Americas, and it has been discovered in Latin America, but also in Israel and South Africa [5,19,21]. In addition, we did not find known mutations in hotspot genes that confer resistance to echinocandins, such as caspofungin, micafungin, and anidulafungin. The genetic/molecular findings correlate with the antifungal susceptibility tests conducted.

As *C. auris* is a multidrug-resistant pathogen that is prone to misidentification by available conventional methods, the true burden of this problem in Nigeria remains unknown, given that most routine clinical laboratory in the country limits *Candida* spp. Identification to just *C. albicans* and non-*C. albicans*, as is the case in many African countries [28]. This is further compounded by the poor availability and accessibility of antifungals in Nigeria (only fluconazole, itraconazole, voriconazole, and amphotericin B deoxycholate are licensed), as three out of four of the patients succumbed to their illnesses and passed away. This calls for the rapid and accurate diagnosis of *C. auris*, followed by genotyping to determine epidemiological relations. This can be achieved by whole genome sequencing, as in this study, or by microsatellite typing [29].

## 5. Conclusions

In this study, we report the first evidence of *C. auris* infections in Nigeria and West Africa, which were further confirmed using whole genome sequencing. All of these isolates were seen to contain substitution mutations in hotspot genes responsible for antifungal resistance to triazoles, which are of serious global health concern, especially in hospitals. The different clades suggest introduction from different places/sources. This work highlights the urgent need for active disease surveillance, especially amongst the “at risk” vulnerable patient population. It also shows how healthcare systems in West Africa can hugely benefit from hospital and academic research collaboration in identifying and diagnosing pathogens that pose a huge threat to human lives.

## Figures and Tables

**Figure 1 jof-08-00787-f001:**
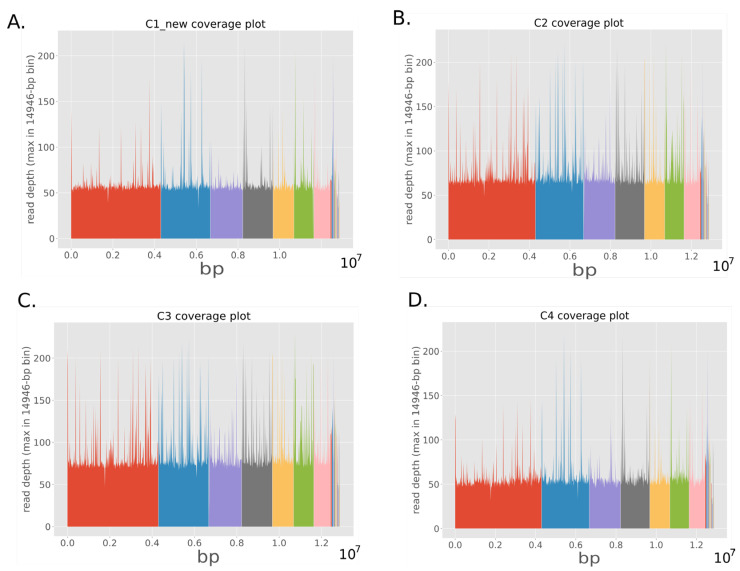
Coverage plot of whole genome sequencing of *C. auris* isolates from case one (**A**), case two (**B**), case three (**C**) and case four (**D**) patients. Each colour represents the different chromosome from 1 to 7 (left to right).

**Figure 2 jof-08-00787-f002:**
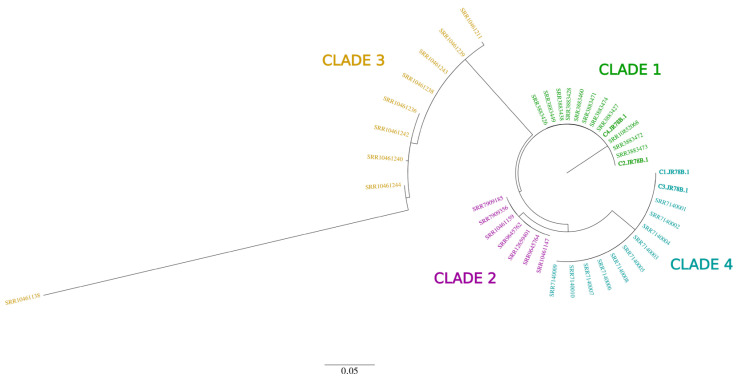
Maximum-likelihood phylogeny of 41 *C. auris* genomes (including four from this study in bold tip labels) showing four distinct phylogeographical clades from all around the world. Genomes from this study cluster with clades 1 and 4.

**Table 1 jof-08-00787-t001:** Demographics and clinical characteristics of reported cases.

Case	1	2	3	4
Date	February, 2021	February, 2021	March, 2021	April, 2021
Gender, age (years)	Male, 60	Female, 67	Male, 74	Female, 48
Underlying disease	Diabetes mellitus, Hypertension	Diabetes mellitus, Hypertension, COVID-19	Prostate cancer	Systemic Lupus Erythematosus (SLE)
Country visited in the previous year	Unknown	Unknown	Dubai, UAE	No travel
Recent intake of broad-spectrum antibiotics	Yes	Yes	Yes	Yes
Vascular surgery	No	No	No	No
Total parenteral nutrition	No	No	No	Yes
Dialysis	Yes	No	No	Yes
Urinary catheterization	Yes	Yes	Yes	Yes
Postoperative drain placement	Yes	No	No	No
Type of invasive candidiasis infection	Bloodstream infection (also isolated from sputum and urine)	Bloodstream infection	Bloodstream (probable skin colonization)	Bloodstream
Antifungal Treatment	Voriconazole	None	None	Fluconazole, voriconazole
Outcome	Death	Death	Discharged	Death

**Table 2 jof-08-00787-t002:** Antifungal susceptibility test results of *Candida auris* isolates.

Antifungal (Minimal Inhibitory Concentration mg/L)							
	Fluconazole	Voriconazole	Posaconazole	Amphotericin B	Caspofungin	Micafungin	Anidulafungin
Tentative MIC breakpoints	≥32	NA *	NA *	≥2	≥2	≥4	≥4
Case 1	16	0.25	0.016	1	0.06	0.063	0.031
Case 2	64	1	0.031	0.5	0.06	0.125	0.25
Case 3	1	0.016	<0.016	0.25	0.06	0.125	0.25
Case 4	32	1	0.031	0.5	0.06	0.125	0.25

* Fluconazole susceptibility is used as a surrogate for second generation triazole susceptibility assessment. NA—Not available.

**Table 3 jof-08-00787-t003:** Detected polymorphisms in hotspot genes that are associated with drug resistance in *C. auris* in the four cases/isolates.

Isolate	Change	Type of Change	Gene ID	Ortholog
Case 1	a1004g | Asn335Ser	snp; missense	CJI97_001156	ERG11
	a1029c | Glu343Asp	snp; missense	CJI97_001156	ERG11
Case 2	a395t | Tyr132Phe	snp; missense	CJI97_001156	ERG11
	ctc374_376ttt | Leu125Phe	complex; missense	CJI97_001156	ERG11
	c4453a | Leu148Ile	snp; missense	CJI97_000983	FKS1
	a2811t | Arg937Ser	snp; missense	CJI97_000983	FKS1
	ca2100ag | Ile701Val	indel; missense	CJI97_000983	FKS1
	a2080g | Ile694Val	snp; missense	CJI97_000983	FKS1
Case 3	a530g | Lys177Arg	snp; missense	CJI97_001156	ERG11
	a1004g | Asn335Ser	snp; missense	CJI97_001156	ERG11
	a1029c | Glu343Asp	snp; missense	CJI97_001156	ERG11
Case 4	ctc374_376ttt | Leu125Phe	complex; missense	CJI97_001156	ERG11
	a395t | Tyr132Phe	snp; missense	CJI97_001156	ERG11
	a117c | Glu39Asp	snp; missense	CJI97_005027	ERG2

## Data Availability

The FASTQ files of isolates generated from this study are availabe in the NCBI Sequence read archive (SRA) under BioProject accession number PRJNA838244.

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
