# Peer review of "Emergence and Genomic Characterization of Multidrug Resistant Candida auris in Nigeria, West Africa"

_jof, 2022, doi:10.3390/jof8080787_

Round 1
Reviewer 1 Report
It would be interesting to discuss the mutations in the FKS1 gene, how they relate to resistance to azoles and amphotericin B, or what significance this would have in relation to the exposure of this strain to echinocandins.
Author Response
Point 1: It would be interesting to discuss the mutations in the FKS1 gene, how they relate to resistance to azoles and amphotericin B, or what significance this would have in relation to the exposure of this strain to echinocandins.
RE: We did not find mutation in the FKS1 gene relevant to resistance. FKS1 mutations have no relation with azole or amphotericin resistance. We only found the most described point mutation in two isolates responsible for azole resistance.
Reviewer 2 Report
The manuscript reports the outbreak of Candida auris in Nigeria. This is the first report of this specie in Nigeria and it is highly valuable to be published. The methods are precisely described and the manuscript is easy to follow for readers, so I would consider this article suitable for publication at Journal of Fungi
Author Response
The manuscript reports the outbreak of Candida auris in Nigeria. This is the first report of this specie in Nigeria and it is highly valuable to be published. The methods are precisely described and the manuscript is easy to follow for readers, so I would consider this article suitable for publication at Journal of Fungi
RE: thank you for your kind remarks.
Reviewer 3 Report
The Manuscript ID: jof-1803403 is entitled "Emergence and Genomic Characterization of Multidrug resistant Candida auris in Nigeria, West Africa". The authors discuss the emergence, the importance of lacking an accurate and rapid identification method in this region, and the subsequent outcomes for the patients. Moreover, in this study, whole-genome sequencing (WGS) of four C. auris isolates from four patients revealed important information on the genetic profile of genes involved in drug resistance. The manuscript is concise and well-written, and it is the first report from the West African region.
Minor corrections:
1. Page 1, line 32: Please change the text format of “Candida” to italics.
2. Page 2, lines 64-65: Please change the sentence font size and style.
3. Page 2, Table 1: in case 3, please add the patient’s age.
4. Page 2, Table 1: for the country visited in the previous year’s data, please use a consistent style for all cases; unknown and none were used.
5. Page 3, line 88: Please change “grew” to “grew”.
6. Page 5, lines 181-185: Please use the same format for antifungal agent names across the whole manuscript. In some instances, the drug name begins with a capital letter, whereas in others, it begins with a lowercase letter.
7. Page 6, Table 3: in the title “C. auris in the four (4) cases/isolates” I guess the (4) is not necessary here.
8. Page 7, lines 221, 223: Please make the gene names "ERG11, ERG2" italics.
9. Page 7, line 233: Please put 2 after VITEK if VITEK 2 was used.
10. Page 7, line 234: Please change the text format of “Candida” to italics.
11. Page 7, line 239: I suggest writing "COIVD-19" in its whole form.
12. Page 7, line 241: “the four (4) major distinct clades” I guess the (4) is not necessary here as well.
13. Page 8, line 250: Please change “Antifungal” to “antifungal”.
Author Response
The Manuscript ID: jof-1803403 is entitled "Emergence and Genomic Characterization of Multidrug resistant Candida auris in Nigeria, West Africa". The authors discuss the emergence, the importance of lacking an accurate and rapid identification method in this region, and the subsequent outcomes for the patients. Moreover, in this study, whole-genome sequencing (WGS) of four C. auris isolates from four patients revealed important information on the genetic profile of genes involved in drug resistance. The manuscript is concise and well-written, and it is the first report from the West African region.
Minor corrections:
1. Page 1, line 32: Please change the text format of “Candida” to italics. RE:Done
2. Page 2, lines 64-65: Please change the sentence font size and style. RE:Done
3. Page 2, Table 1: in case 3, please add the patient’s age. RE:Done
4. Page 2, Table 1: for the country visited in the previous year’s data, please use a consistent style for all cases; unknown and none were used
RE: we changed none to no travel.
5. Page 3, line 88: Please change “grew” to “grew”. RE:Done
6. Page 5, lines 181-185: Please use the same format for antifungal agent names across the whole manuscript. In some instances, the drug name begins with a capital letter, whereas in others, it begins with a lowercase letter. RE:Done
7. Page 6, Table 3: in the title “C. auris in the four (4) cases/isolates” I guess the (4) is not necessary here. RE:Done
8. Page 7, lines 221, 223: Please make the gene names "ERG11, ERG2" italics. RE:Done
9. Page 7, line 233: Please put 2 after VITEK if VITEK 2 was used. RE:Done
10. Page 7, line 234: Please change the text format of “Candida” to italics. RE:Done
11. Page 7, line 239: I suggest writing "COIVD-19" in its whole form. RE:Done
12. Page 7, line 241: “the four (4) major distinct clades” I guess the (4) is not necessary here as well. RE: corrected
13. Page 8, line 250: Please change “Antifungal” to “antifungal”. RE:Done
Reviewer 4 Report
This study reports first Candida auris isolates from Nigeria and Western Africa. The isolates were discovered in a short period of time and belonged to different clades which mostly emerged in different geographical regions (Clade 1 and Clade 4). It is interesting that Clade 3, which emerged in South Africa was not detected in this study. This might underline the need for ongoing surveillance of this extraordinary pathogen.
The article gives information on 4 cases. Although all isolates were obtained from blood, the clinical evaluation of Case 3 suggested colonization. Was blood cultured obtained through catheter? Any evidence of skin colonization? Was the urinary isolate identified to species level? If possible, please add information.
In lines 183-185 antifungal susceptibility results were summarized as “Results exhibited high minimum inhibitory concentrations (MIC) to triazoles and amphotericin B in cases 1, 2 and 3 and susceptibility to fluconazole in case 4 (Table 2).” However, in Table 2, fluconazole susceptible isolate is from Case 1. Please clarify.
In line 258, molecular identification is recommended for diagnosis of C.auris infections. Molecular methods are expensive and time consuming. This study successfully identified C.auris to species level using a commercial biochemical system (VITEK2) which points out the evolving databases. In addition, MALDITOF-MS might be a cheaper, faster and reliable identification method, if available. Perhaps the authors wanted to emphasize the need for molecular epidemiological testing, which needs to be clarified.
Ethics approval should be added as the study includes patient data.
Minor typos and suggestions:
Line 19, 88, 89, 90, 114, …: “Candida spp.” instead of “Candida sp” OR “Candida sp.” for all including line 88
Line 88: “Candida spp.” should be “Candida spp.” OR “Candida sp.”
Line 32, 57: “Candida” should be in italics.
Line 59: “approximately” or “approx.” instead of “aprox”
Line 66: “Table 1 below is a summary of patients…”
Lines 75,76, 86, 87, 93, 94, 106, 182, 222 …: names of antimicrobials and other drugs should not begin with capital letters.
Line 82: “Chest X-ray performed on the 9th day of ICU admission”
Line 88: “grew” should not be italicized.
Line 106: Please check spelling for “bendroflumethiide”
Line 123: “… bladder cancer prostrate …” Please clarify if the patient has bladder cancer, prostate cancer or both? Table 1 states only as prostate cancer.
Line 132: “Blood culture yielded C. auris…” instead of “Blood culture yield of C. auris…”
Line 144: “Blood culture grew C. auris” instead of “Blood culture done grew C. auris”
Line 145: Please check spelling of voriconazole.
Lines 150, 153, 156, 157…: Names of commercial kits should be capitalized such as “Plant/Fungi DNA Isolation Kit”, “Nextera DNA Flex Preparation Kit” etc.
Line 167: All gene names should be italicized.
Line 182 suggestion: “…were performed for fluconazole, voriconazole…” instead of “were performed using: Fluconazole, Voriconazole”
Lines 25, 49, 53, Table 1, 227: “colonization”, “colonized” and Line 132 “colonisation”. Please prefer one of the spellings.
Table 1 suggestion: “Prostate cancer” instead of “Cancer of prostate”
Table 2: Please add “**” to explanations, such as: “**NA: Not available”.
Table 3: The text states that 5 non-synonymous mutations were detected in all isolates (line 202) but not all were listed in Table 3. For example Y132 F (Tyr132Phe) was given in Case 2 and Case 4 only. N335S (Asn335Ser) and E343D (Glu343Asp) was given in Case 1 and Case 3 only. Please check and clarify.
Author Response
This study reports first Candida auris isolates from Nigeria and Western Africa. The isolates were discovered in a short period of time and belonged to different clades which mostly emerged in different geographical regions (Clade 1 and Clade 4). It is interesting that Clade 3, which emerged in South Africa was not detected in this study. This might underline the need for ongoing surveillance of this extraordinary pathogen.
The article gives information on 4 cases. Although all isolates were obtained from blood, the clinical evaluation of Case 3 suggested colonization. Was blood cultured obtained through catheter? Any evidence of skin colonization? Was the urinary isolate identified to species level? If possible, please add information.
RE:Sample and isolate was shipped to our laboratory from a different city, the doctor who provided clinical information could not ascertain if it was via catheter, but this is most likely since most ICU patients in our setting have a central venous catheters (CVC) in-situ. The urinary isolate was not identified to species level as shown in the paragraph but we confirmed the isolate from blood to be C. auris.
In lines 183-185 antifungal susceptibility results were summarized as “Results exhibited high minimum inhibitory concentrations (MIC) to triazoles and amphotericin B in cases 1, 2 and 3 and susceptibility to fluconazole in case 4 (Table 2).” However, in Table 2, fluconazole susceptible isolate is from Case 1. Please clarify.
RE: thank you. Your remark is correct. We corrected this mistake and changed case three to fluconazole susceptible as there was a mix up with the case IDs from the susceptibility test. In addition we removed the suggestion that amphotericin was resistant.
In line 258, molecular identification is recommended for diagnosis of C.auris infections. Molecular methods are expensive and time consuming. This study successfully identified C.auris to species level using a commercial biochemical system (VITEK2) which points out the evolving databases. In addition, MALDITOF-MS might be a cheaper, faster and reliable identification method, if available. Perhaps the authors wanted to emphasize the need for molecular epidemiological testing, which needs to be clarified.
RE: You are correct. That molecular identification is necessary with updated VITEK cards and Malditof. We menat to emphasioze molecular genotyping to determine the epidemiology. This has now been changed in the text.
Ethics approval should be added as the study includes patient data.
RE: Ethics approval now included
Minor typos and suggestions:
Line 19, 88, 89, 90, 114, …: “Candida spp.” instead of “Candida sp” OR “Candida sp.” for all including line 88. RE:Done
Line 88: “Candida spp.” should be “Candida spp.” OR “Candida sp.” RE:Done
Line 32, 57: “Candida” should be in italics. RE:Done
Line 59: “approximately” or “approx.” instead of “aprox” RE:Done
Line 66: “Table 1 below is a summary of patients…” RE:Done
Lines 75,76, 86, 87, 93, 94, 106, 182, 222 …: names of antimicrobials and other drugs should not begin with capital letters. RE:Done
Line 82: “Chest X-ray performed on the 9th day of ICU admission” RE:Done
Line 88: “grew” should not be italicized. RE:Done
Line 106: Please check spelling for “bendroflumethiide” RE:Done
Line 123: “… bladder cancer prostrate …” Please clarify if the patient has bladder cancer, prostate cancer or both? Table 1 states only as prostate cancer. RE: Done
Line 132: “Blood culture yielded C. auris…” instead of “Blood culture yield of C. auris…” RE:Done
Line 144: “Blood culture grew C. auris” instead of “Blood culture done grew C. auris” RE:Done
Line 145: Please check spelling of voriconazole. RE:Done
Lines 150, 153, 156, 157…: Names of commercial kits should be capitalized such as “Plant/Fungi DNA Isolation Kit”, “Nextera DNA Flex Preparation Kit” etc. RE: Done
Line 167: All gene names should be italicized. RE:Done
Line 182 suggestion: “…were performed for fluconazole, voriconazole…” instead of “were performed using: Fluconazole, Voriconazole” RE:Done
Lines 25, 49, 53, Table 1, 227: “colonization”, “colonized” and Line 132 “colonisation”. Please prefer one of the spellings. RE:Done
Table 1 suggestion: “Prostate cancer” instead of “Cancer of prostate” RE:Done
Table 2: Please add “**” to explanations, such as: “**NA: Not available”. RE:Done
Table 3: The text states that 5 non-synonymous mutations were detected in all isolates (line 202) but not all were listed in Table 3. For example Y132 F (Tyr132Phe) was given in Case 2 and Case 4 only. N335S (Asn335Ser) and E343D (Glu343Asp) was given in Case 1 and Case 3 only. Please check and clarify. RE: The text has been reworded to avoid confusion and now reads as “Non-synonymous mutations were detected in the ERG11 gene of all isolates, with case 2 having additional non-synonymous mutations...”